# Estimation of the morbidity and mortality of congenital Chagas disease: A systematic review and meta-analysis

**Sarah Matthews**[1], **Ayzsa Tannis**[1]*, **Karl Philipp Puchner**[2], **Maria Elena Bottazzi**[3,4], **Maria Luisa Cafferata**[5,6], **Daniel Comandé**[5], **Pierre Buekens**[1]

**1** Tulane University School of Public Health and Tropical Medicine, New Orleans, Louisiana, United States of America, **2** German Leprosy and TB Relief Association, Würzburg, Germany, **3** National School of Tropical Medicine, Department of Pediatrics, Department of Molecular Virology and Microbiology, Baylor College of Medicine, Houston, TX, United States of America, **4** Texas Children's Hospital Center for Vaccine Development, Houston, Texas, United States of America, **5** Instituto de Efectividad Clínica y Sanitaria, Buenos Aires, Argentina, **6** Unidad de Investigación Clínica y Epidemiológica Montevideo (UNICEM), Montevideo, Uruguay

☯ These authors contributed equally to this work.
* atannis@tulane.edu

**Data Availability Statement:** All relevant data are within the manuscript and its Supporting Information files.

## Abstract

Chagas disease is caused by the parasite *Trypanosoma cruzi* which can be transmitted from mother to baby during pregnancy. There is no consensus on the proportion of infected infants with clinical signs of congenital Chagas disease (cCD). The objective of this systematic review is to determine the burden of cCD. Articles from journal inception to 2020 reporting morbidity and mortality associated with cCD were retrieved from academic search databases. Observational studies, randomized-control trials, and studies of babies diagnosed with cCD were included. Studies were excluded if they were case reports or series, without original data, case-control without cCD incidence estimates, and/or did not report number of participants. Two reviewers screened articles for inclusion. To determine pooled proportion of infants with cCD with clinical signs, individual clinical signs, and case-fatality, random effects meta-analysis was performed. We identified 4,531 records and reviewed 4,301, including 47 articles in the narrative summary and analysis. Twenty-eight percent of cCD infants showed clinical signs (95% confidence interval (CI) = 19.0%, 38.5%) and 2.2% of infants died (95% CI = 1.3%, 3.5%). The proportion of infected infants with hepatosplenomegaly was 12.5%, preterm birth 6.0%, low birth weight 5.8%, anemia 4.9%, and jaundice 4.7%. Although most studies did not include a comparison group of non-infected infants, the proportion of infants with cCD with clinical signs at birth are comparable to those with congenital toxoplasmosis (10.0%-30.0%) and congenital cytomegalovirus (10.0%-15.0%). We conclude that cCD burden appears significant, but more studies comparing infected mother-infant dyads to non-infected ones are needed to determine an association of this burden to cCD.

**Funding:** The authors received no specific funding for this work.

**Competing interests:** The authors have declared that no competing interests exist.

## Author summary

Chagas disease is caused by the parasite *Trypanosoma cruzi*, which can be passed from mother to infant. It is estimated that one million women of reproductive age are infected with *Trypanosoma cruzi*. Prior to our work, the proportion of infants with congenital Chagas disease (cCD) presenting with clinical signs was unknown. After systematically searching for and identifying studies that collected information on infants with cCD, we summarized and analyzed 47 studies. Our pooled analysis of these studies estimated that 28.3% of infants with cCD showed clinical signs and 2.2% died. Prior work has shown that transmission of *T. cruzi* from mother to child occurs in 5% of cases. Other studies have shown that this transmission is preventable through treatment of women prior to conception, and infants can be cured if shown to be infected at birth. Our estimated proportion of 28.3% of infants diagnosed with cCD at birth presenting with clinical signs are comparable to infants diagnosed with congenital toxoplasmosis presenting with clinical signs (10.0%-30.0%) and congenital cytomegalovirus (10.0%-15.0%). More studies comparing infected mother-infant dyads to non-infected mother-infant dyads are needed to determine an association of this burden to cCD.

## Introduction

### Background

Chagas disease, caused by the protozoan parasite *Trypanosoma cruzi*, is estimated to infect 6.5 million globally, including 1.7 million women of reproductive age [1, 2]. As of 2019, an estimated 172,000 additional people were infected, and 52,000 of these were women of reproductive age [1]. *Trypanosoma cruzi* is primarily transmitted when the triatomine insect vector transfers the parasite after biting and defecating on its host through its infected feces entering via bite wound or mucosal membrane [2]. However, it can also be transmitted through blood transfusion, organ transplant, via oral consumption of contaminated food or beverage, and through vertical transmission from mother to infant during pregnancy [3, 4].

Vertical transmission of *T. cruzi*, or congenital Chagas disease (cCD), occurs in an estimated 4.7% of infants born to infected mothers, increasing to 5.0% in endemic countries [3]. *Trypanosoma cruzi* infected infants may present with severe morbidity at birth and be at a higher risk of mortality. If left untreated, infants can develop chronic Chagas disease later in life [4].

### Clinical signs of congenital Chagas disease

Clinical signs in infants with cCD range from mild to severe. Clinical signs attributable to cCD include low Apgar score (<7 at 1 minute and/or at 5 minutes) [5], premature rupture of membranes [6], preterm birth, low birth weight [7], intra-uterine growth restriction [8], small for gestational age [9], neonatal intensive care unit (ICU) admission [10], hepatomegaly, splenomegaly, respiratory distress syndrome, certain neurologic signs, anasarca, petechiae, abnormal electrocardiographic findings, anemia, meningoencephalitis, myocarditis, congestive heart failure, digestive and/or central nervous system lesions, parasites in various tissues [2], subependymal hemorrhage [11], and cardiomegaly [6]. Mortality attributed to cCD is associated with severe morbidity, including meningoencephalitis and myocarditis [2].

### Clinical pathway

Infants exposed in-utero to *Trypanosoma cruzi* are susceptible to congenital transmission [12]. Screening programs to diagnose and treat pregnant women and congenital *T. cruzi* infection

in infants have been implemented in endemic countries and countries with large migrant populations from endemic regions since the early 1990s [13, 14]. Per guidelines published by the Pan American Health Organization in 2019, the gold standard for diagnosing acute and chronic infection uses at least two conventional serological tests (e.g., indirect hemagglutination assay, indirect immunofluorescence assay, ELISA) [12, 15]. Other tests, such as molecular tests and rapid diagnostic tests, can also confirm infection but are only recommended to complement or confirm aforementioned assays [12,15].

Confirmation of cCD in infants born to *T. cruzi* infected mothers occurs at birth or in the first weeks afterward by viewing parasites in an umbilical cord blood sample or venous infant blood, or after 8–10 months when maternal antibodies have waned using serological assays to confirm infant *T. cruzi* IgG antibodies [4, 12]. Gold standard diagnosis of cCD requires, at birth, parasitological examination using microhematocrit or microStrout testing methods, and if negative, repeated examination one month later, and at 10 months two serological tests [12]. Molecular method of diagnosis using PCR can detect infection early on but are not part of the gold standard diagnosis given lack of standardization, low and often fluctuating parasitemia in patients with chronic Chagas disease, and lack of quality control programs [12, 15, 16].

Evidence shows that if women are treated for Chagas disease before pregnancy, future congenital transmission of *T. cruzi* is preventable [4]. Treatment during pregnancy is not recommended given unknown effects of antiparasitic drugs on prenatal development. Treatment of *T. cruzi* infected infants with benznidazole or nifurtimox is effective when administered within the first year of life [17].

## Rationale

In 2010, it was estimated that between 158,000 to 214,000 infants were born to *T. cruzi* infected mothers in endemic countries, of which 8,000 to 10,700 would be congenitally infected [18]. Around one-fifth of annual new Chagas cases are estimated to be congenital infections [18]. The Global Burden of Disease project used data from vital registration databases, surveillance, surveys/census, and other population-based sources to estimate the burden of Chagas disease among neonates and infants, including number of deaths, disability-adjusted life years, years lived with disability, and years of life lost [19]. However, it is likely that the data used to estimate the burden of Chagas disease are incomplete given issues in diagnosing cCD, including low sensitivity of parasitological screening at birth, and loss to follow-up with serological screening 8 to10 months postpartum [18]. Given this, there is no accurate burden estimate for cCD and no consensus on how many infants have clinical signs [2]. The objective of this systematic review is to determine the morbidity and mortality of cCD.

## Methods

A systematic review and meta-analysis were performed according to the guidelines of the Meta-Analysis of Observational Studies in Epidemiology (MOOSE) and the Preferred Reporting Items for Systematic reviews and Meta-Analysis (PRISMA) [20, 21]. The protocol was registered on PROSPERO (CRD42020165987) [22].

### Criteria for considering studies

**Types of studies.**   Articles that reported original data of morbidity or mortality associated with cCD were considered, including observational studies and randomized control trials. Studies excluded were case reports and series, studies not including original data, case-control studies without neonatal incidence estimates of cCD, and studies not reporting the number of infected neonates.

**Types of participants.**   Studies about diagnosed neonates and infants with cCD were included.

**Types of outcomes.**   Mortality was defined as the recorded death of a *T. cruzi* congenitally infected fetus or infant. Morbidity was defined as any adverse outcome presenting in a *T. cruzi* congenitally infected infant, with all clinical signs extracted available in the **S1 File.** Mortality causes included stillbirth, miscarriage, abortion, intrauterine death, and fetal death.

## Search strategy

A medical librarian developed and applied a comprehensive and sensitive search strategy (available in **S2 File**) using terms related to cCD in PubMed, EMBASE, CINAHL, LILACS, and Academic Search databases. No language restrictions were applied, and grey literature was not searched.

## Data collection and analysis

**Selection of studies.**   Authors AT and SM independently screened article titles and abstracts and then the remaining full text articles for eligibility. All disagreements were resolved by discussion and, if necessary, a third author (KP) was consulted as an arbitrator. Covidence systematic review software was used to facilitate the screening process [23]. For duplicate articles, the one with the largest sample size was included. The decision-making algorithm consideration is available in **S3 File.**

## Data extraction and management

Authors AT and SM independently extracted data using a form designed and piloted with studies a priori. Extracted data included study, maternal, and infant characteristics, diagnostic information for mothers and infants, and morbidity and mortality of congenital cases. A summary of extracted data can be found in **S4 File** and the data extraction form in **S1 Dataset.**

Data extraction discrepancies were resolved by discussion and, if necessary, a third author (KP) was consulted. The inter-observer reviewer agreement for full text screening was assessed using the Kappa statistic.

**Assessment of risk of bias.**   A risk of bias assessment tool was developed through adaptation of the NIH Study Quality Assessment Tools and the Strengthening the Reporting of Observational studies in Epidemiology (STROBE) checklist of essential items [24, 25]. Authors AT and SM piloted the tool on five studies, subsequently adapted the tool and then independently assessed risk of bias of the included studies (**S5 File**). Six domains were considered: 1) participant selection methods, 2) exposure and outcome variable measurement, 3) confounding control methods, 4) reporting of results, 5) statistical methods, and 6) declaration of conflict and ethical statements. Two algorithms were developed to summarize within-domain and summary risk of bias (**S5 File**).

**Statistical analysis and data synthesis.**   Included study frequencies of congenital transmission, cCD clinical signs, mortality causes (including those not originally listed in **S1 File**), infant mortality and/or case-fatality rates, and proportion of cCD cases with and without clinical signs were narratively summarized.

A meta-analysis of proportions was performed to estimate the pooled proportion of fetuses and infants with cCD with clinical signs. The Freeman-Tukey double arcsine method was used to account for overdispersion of proportions and stabilize the variance [26, 27]. Stuart-Ord inverse variance weight was applied to transformed proportions, avoiding underestimation of true variance using its conservative weight [28]. The pooled proportion and its 95% confidence interval (CI) were estimated using the DerSimonian-Laird random effects model to take into

consideration the high likelihood of between-study heterogeneity. Results were quantified and represented in a forest plot [28, 29]. The proportion of cCD cases with clinical signs was defined as the number of infants with cCD displaying clinical sign(s) and/or death divided by the total number of infants with cCD. We also performed a meta-analysis of proportions for the pooled proportion of death due to cCD. If a study reported cCD clinical signs and/or mortality frequency but did not provide a frequency for every outcome outlined in **S1 File** and/or death, missing values were assumed to be non-events and a value of 0 was imputed [30]. All analyses were performed using SAS Version 9.4, Stats-Direct, and StataIC 12 software.

**Assessment of heterogeneity.** The $I^2$ statistic was calculated to measure the proportion of total variability attributable to heterogeneity between studies [31]. Three subgroup analyses defined a priori were performed by: cCD diagnostic method, geographic region, and individual clinical sign displayed in fetus/infant. Studies were excluded for the subgroup analysis of clinical sign frequency if no clear definition of each clinical sign displayed in individual infants was reported. A subgroup of co-infection with other non-Chagas related infections was planned; however, data was insufficient. A post-hoc subgroup analysis was conducted by year (s) study data was collected; an additional analysis by *T.cruzi* discrete typing unit (DTU) was planned, but data were insufficient. Detailed results are described in **S6 File.**

**Sensitivity analyses.** Two sensitivity analyses were conducted to assess the potential effect review decisions held on robustness of results. These analyses were to exclude studies with high risk of bias and exclude studies where Chagas disease gold standard diagnosis of the mother was not employed [15]. An ad-hoc sensitivity analysis was performed using the Miller back-transformation [32] for the primary meta-analyses of proportion of cCD morbidity and mortality. Detailed results are described in **S6 File.**

**Assessment of publication bias.** The effect of publication bias was evaluated for all analyses using Egger's statistical test to determine asymmetry of the funnel plot [33].

## Results

A total of 4,531 records were identified through database search, 4,301 were screened based on title and abstract, 293 full text articles assessed for eligibility, and 47 articles were included for narrative summary and meta-analysis.

### Narrative summary

Study publication year ranged from 1962 to 2019, with 18 studies published before 2000, 10 between 2000 to 2010, and 18 between 2011 and 2019. Data collection timeframe was reported in 46 studies (with some overlap between timeframes), with data collection conducted before 2000 in 21 studies, between 2000 and 2010 in 20 studies, and after 2010 in eight studies. Study duration ranged from under one year to 15 years, with six studies under one year, 31 studies one to four years, and eight studies five to 15 years, with two studies missing data on this factor. Twelve studies were conducted in Europe and 35 in the Latin American region (Mexico, Central and South America). Most studies (n = 38) were conducted in urban/semi-urban hospital(s), with one conducted in a rural hospital, four in both rural and urban hospitals, one conducted in primary care institutions, and three with missing information. With regard to study design, three were case-control, 13 cross-sectional, 28 prospective cohorts, two retrospective cohorts, and one a mixed cohort. Study population size varied from eight to 4,355 infants. Fifteen studies had less than 100 infants, 14 had between 100 and 999 infants, and 10 had over 1,000 infants, with five studies missing data. Twenty-eight studies used gold standard diagnosis for mothers, 16 used an alternative, and three studies did not provide information. **Table 1** summarizes all study characteristics.

**Table 1. Characteristics of included articles and their study population.**

| | Article characteristics | | | | Maternal characteristics | | | Infant characteristics | | | | |
|---|---|---|---|---|---|---|---|---|---|---|---|---|
| Article | Country-city | Study period | Study setting | Study design | # | # Infected | Method diagnosis | # | # Infected | Method diagnosis | # Without clinical signs (%) | # With clinical signs (%) |
| Apt 2013 [34] | Chile-Salamanca Chile-Illapel Chile-Los Vilos Chile-Canela | 2005–2009 | Rural hospitals | Prospective cohort | 4831 | 147 | Gold | 147 | 6 | Other | 3 (50.0) | 3 (50.0) |
| Arcavi 1993 [35] | Argentina—CABA | 01/1990–02/1991 | Urban/semiurban hospital | Prospective cohort | 729 | 62 | Gold | 62 | 2 | Other | 2 (100.0) | 0 (0.0) |
| Azogue 1991 [36] | Bolivia—Santa Cruz | 03/1988–12/1989 | Urban hospital | Case control | 760 | 410 | Other | 820 | 78 | Gold | 57(73.0) | 21(27.0) |
| Bahamonde 2002 [37] | Chile—Antofagasta | 11/1996–10/1997 | Urban/semiurban hospital | Prospective cohort | Not specified | Not specified | Gold | 1987 | 5 | Other | 5 (100.0) | 0 (0.0) |
| Barona—Vilar 2012 [38] | Spain—Valencia | 2009–2010 | Urban/semiurban hospitals | Cross sectional | 1975 | 226 | Gold | Not specified | 8 | Gold | 7 (87.5) | 1 (12.5) |
| Barousse 1978 [39] | Argentina—CABA | 07/1976–07/1977 | Not specified | Prospective cohort | 4220 | 186 | Other | 186 | 1 | Other | 0 (0.0) | 1 (100.0) |
| Basile 2019 [40] | Spain—Catalonia | 2010–2015 | Mixed urban/rural hospitals | Prospective cohort | 33469 | 818 | Gold | 812 | 28 | Gold | 24 (85.7) | 4 (14.3) |
| Bern 2009 [41] | Bolivia—Santa Cruz | 11/2006–06/2007 | Urban/semiurban hospital | Prospective cohort | 530 | 154 | Gold | 138 | 10 *7 with data | Gold | 4 (57.1) | 3 (42.9) |
| Bisio 2011 [42] | Argentina—CABA | 2002–2007 | Urban/semiurban hospital | Prospective cohort | 104 | 104 | Gold | 83 | 3 | Gold | 3 (100.0) | 0 (0.0) |
| Bittencourt 1985 [43] | Brazil—Salvador | 01/1981–08/1982 | Urban/semiurban hospitals | Prospective cohort | 2651 | 226 | Gold | 186 | 3 | Not specified | 1 (33.3) | 2 (66.7) |
| Buekens 2018 [44] | Argentina—San Miguel de Tucuman Mexico—Merida Honduras—Santa Barbara Honduras—Intibuca | 2011–2013 | Urban/semiurban hospitals | Prospective cohort | 28145 | 347 | Gold | 503 | 11 | Gold | 7 (63.6) | 4 (36.4) |
| Cardoso 2012 [45] | Mexico—Santiago Pinotepa Nacional Mexico—Potchutla Mexico—Guadalajara Mexico—Mexico City | 09/2006–06/2008 | Urban/semiurban hospitals | Prospective cohort | 1448 | 106 | Other | 106 | 15 | Other | 14 (93.3) | 1 (6.7) |
| Castillo 1984 [46] | Chile—Antofagasta Chile—Calama | 08/1983–06/1984 | Urban/semiurban hospitals | Cross sectional | 1952 | 35 | Other | 1961 | 31 | Other | 29 (93.6) | 2 (6.5) |
| Contreras 1999 [47] | Argentina—General Guemes | 08/1996–12/1996 | Not specified | Cross sectional | 276 | 34 | Gold | 34 | 3 | Other | 3 (100.0) | 0 (0.0) |

(*Continued*)

**Table 1.** (Continued)

| Article characteristics | | | | | Maternal characteristics | | | Infant characteristics | | | | |
|---|---|---|---|---|---|---|---|---|---|---|---|---|
| Article | Country-city | Study period | Study setting | Study design | # | # Infected | Method diagnosis | # | # Infected | Method diagnosis | # Without clinical signs (%) | # With clinical signs (%) |
| Cucunuba 2012 [48] | Colombia—Arauca Colombia—Boyaca Colombia—Casanare Colombia—Meta Colombia—Santander | 01/2010–12/2011 | Other | Cross sectional | 4417 | 119 | Gold | 47 | 5 | Other | 5 (100.0) | 0 (0.0) |
| De Rissio 2010 [49] | Argentina—CABA Argentina–Buenos Aires Metropolitan Area | 10/1994–12/2004 | Urban/ semiurban hospital | Prospective cohort | 6204 | 265 | Gold | 4355 | 267 | Gold | 267 (100.0) | 0 (0.0) |
| Flores—Chavez 2011 [50] | Spain—Madrid | 01/2008–12/2010 | Urban/ semiurban hospitals | Retrospective cohort | 3839 | 152 | Other | 152 | 4 | Other | 3 (75.0) | 1 (25.0) |
| Francisco—Gonzáles 2018 [51] | Spain—Madrid | 01/2012–09/2016 | Urban/ semiurban hospitals | Retrospective cohort | 122 | 122 | Gold | 125 | 3 | Other | 2 (66.7) | 1 (33.3) |
| Freilij 1995 [52] | Argentina—CABA | 1987–1993 | Urban/ semiurban hospital | Mixed cohort | Not specified | 1116 | Not specified | 1118[a] | 71 | Other | 46 (64.8) | 25 (35.2) |
| Fumado 2014 [53] | Spain—Barcelona | 03/2003–09/2008 | Urban/ semiurban hospital | Prospective cohort | Not specified | Not specified | Not specified | 72[b] | 5 | Other | 5 (100.0) | 0 (0.0) |
| Giménez 2010 [54] | Spain—Valencia | 06/2007–10/2009 | Urban/ semiurban hospital | Prospective cohort | 574 | 35 | Gold | 35 | 3 | Other | 2 (66.7) | 1 (33.3) |
| Iglesias 1985 [55] | Chile—Santiago | 01/1985–06/1985 | Urban/ semiurban hospital | Cross sectional | 1000 | 11 | Other | 1000 | 9 | Not specified | 9 (100.0) | 0 (0.0) |
| Mallimaci 2010 [56] | Argentina—Ushuaia | 02/2001–12/2002 | Urban/ semiurban hospital | Prospective cohort | 61 | 61 | Gold | 68 | 3 | Gold | 3 (100.0) | 0 (0.0) |
| Martínez de Tejada 2009 [57] | Switzerland—Geneva | 2008 | Urban/ semiurban hospitals | Prospective cohort | 305 | 6 | Other | 8 | 2 | Other | 1 (50.0) | 1 (50.0) |
| Mayer 2010 [58] | Argentina—CABA | 2000–2005 | Urban/ semiurban hospital | Case-control | Not specified | Not specified | Gold | 1058 | 18 | Other | 9 (50.0) | 9 (50.0) |
| Mendoza 2014 [59] | Spain—Barcelona | 07/2010–12/2013 | Urban/ semiurban hospital | Prospective cohort | 1717 | 81 | Other | 81 | 5 | Gold | 5 (100.0) | 0 (0.0) |
| Mendoza 1983 [60] | Chile—Copiapo | 10/1982–06/1983 | Urban/ semiurban hospital | Cross sectional | 869 | 31 | Other | 875 | 30 | Other | 30 (100.0) | 0 (0.0) |
| Messenger 2017 [61] | Bolivia—Santa Cruz de la Sierra Bolivia—Camiri | 2010–2014 | Urban/ semiurban hospitals | Prospective cohort | 1851 | 476 | Gold | 487 | 38 | Gold | 27 (71.1) | 11 (29.0) |

(*Continued*)

**Table 1.** (Continued)

| Article | Country-city | Study period | Study setting | Study design | # | # Infected | Method diagnosis | # | # Infected | Method diagnosis | # Without clinical signs (%) | # With clinical signs (%) |
|---|---|---|---|---|---|---|---|---|---|---|---|---|
| | | | | | | Article characteristics | | Maternal characteristics | | | Infant characteristics | |
| Munoz 2009 [62] | Spain—Barcelona | 03/2005–09/2007 | Urban/semiurban hospitals | Prospective cohort | 1350 | 46 | Other | 46 | 3 | Gold | 3 (100.0) | 0 (0.0) |
| Munoz 1982 [63] | Chile—Santiago | 05/1979–11/1979 | Urban/semiurban hospital | Prospective cohort | 402 | 11 | Other | 402 | 2 | Other | 0 (0.0) | 2 (100.0) |
| Murcia 2017 [64] | Spain—Murcia | 01/2007–05/2016 | Urban/semiurban hospital | Case-control | 144 | 144 | Gold | 160 | 16 | Gold | 13 (81.3) | 3 (18.8) |
| Nisida 1999 [65] | Brazil—Sao Paulo City | Not specified | Urban/semiurban hospitals | Cross sectional | 57 | 57 | Gold | 58 | 4 | Other | 0 (0.0) | 4 (100.0) |
| Oritz 2012 [66] | Chile—Region IV Choapa | 2006–2010 | Not specified | Prospective cohort | 110 | 110 | Gold | 100 | 3 | Other | 3 (100.0) | 0 (0.0) |
| Otero 2012 [67] | Spain—Barcelona | 04/2008–05/2010 | Urban/semiurban hospital | Prospective cohort | 633 | 22 | Gold | 22 | 1 | Gold | 0 (0.0) | 1 (100.0) |
| Rodari 2018 [68] | Italy—Bergamo | 01/2014–12/2016 | Mixed urban/rural hospitals | Prospective cohort | 376 | 28 | Gold | 29 | 1 | Gold | 0 (0.0) | 1 (100.0) |
| Rubio 1962 [69] | Chile—Santiago | 1959 | Urban/semiurban hospitals | Cross sectional | 100 | 3 | Other | 50 | 1 | Other | 0 (0.0) | 1 (100.0) |
| Salas 2007 [70] | Bolivia—Yacuiba | 05/2003–09/2004* | Urban/semiurban hospital | Prospective cohort | 2712 | 1144 | Gold | 2742 | 58 | Other | 43 (74.1) | 15 (25.9) |
| Sasagawa 2015 [71] | El Salvador—Santa Isabel Ishuatan El Salvador—Armenia El Salvador—San Antonio del Monte El Salvador—Guaymango | 03/2009–02/2010 09/2009–05/2010 | Mixed urban/rural hospitals | Prospective cohort | 943 | 36 | Other | 36 | 1 | Other | 1 (100.0) | 0 (0.0) |
| Sosa—Estani 2009 [72] | Argentina—Formosa | 01/2005–06/2006* | Urban/semiurban hospital | Prospective cohort | 271 | 79 | Gold | 108 | 8 | Other | 6 (75.0) | 2 (25.0) |
| Streiger 1995 [73] | Argentina—Santa Fe | 1976–1991 | Urban/semiurban hospitals | Prospective cohort | 6123 | Not specified | Gold | 341 | 9 | Other | 3 (33.3) | 6 (66.7) |
| Tello 1982 [74] | Chile—Santiago | 05/1981–07/1982 | Urban/semiurban hospital | Cross sectional | 1000 | 27 | Other | 100 | 3 | Other | 3 (100.0) | 0 (0.0) |
| Torrico 2004 [6] | Bolivia—Cochabamba | 11/1992–07/1994 02/1999–11/2001 | Urban/semiurban hospital | Prospective cohort | Not specified | Not specified | Gold | Not specified | 71 | Other | 35 (49.3) | 36 (50.7) |

(*Continued*)

**Table 1.** (Continued)

| Article | Country-city | Study period | Study setting | Study design | # | # Infected | Method diagnosis | # | # Infected | Method diagnosis | # Without clinical signs (%) | # With clinical signs (%) |
|---|---|---|---|---|---|---|---|---|---|---|---|---|
| | *Article characteristics* | | | | *Maternal characteristics* | | | *Infant characteristics* | | | | |
| Valenzuela 1984 [75] | Chile— Rancagua Chile—San Fernando Chile—Santa Cruz | 04/1984– 12/1984 | Mixed urban/rural hospitals | Cross sectional | 2135 | 23 | Other | 2146 | 11 | Other | 7 (63.6) | 4 (36.4) |
| Valperga 1992 [76] | Argentina— San Miguel de Tucuman | 05/1990– 06/1991 | Urban/ semiurban hospitals | Cross sectional | 1434 | Not specified | Not specified | 1496 | 4 | Other | 1(25.0) | 3 (75.0) |
| Vicco 2016 [77] | Bolivia— Yacuiba | Not specified | Urban/ semiurban hospital | Cross sectional | 183 | 64 | Gold | 172 | 4 | Other | 4 (100.0) | 0 (0.0) |
| Villablanca 1984 [78] | Chile—San Felipe Chile—Los Andes | 04/1983– 12/1984 | Urban/ semiurban hospitals | Cross sectional | 2099 | 62 | Other | 2104 | 61 | Other | 36 (59.0) | 25 (41.0) |
| Zaidenberg 1993 [79] | Argentina— Salta | 1981– 1985 | Urban/ semiurban hospital | Cross sectional | 937 | 149 | Gold | 929 | 12 | Other | 0 (0.0) | 12 (100.0) |

[a]Children recruited at various ages: 733 <6 months, 532 >6 months

[b]Only reporting the number of children in the study under the age of 1 year.

*Indicates a study whose follow-up ended past the end date: Salas 2007 [70] ended follow-up in 2005, and Sosa-Estani 2009 [72] ended follow-up in 2007.

The number of infants with cCD in studies ranged from one to 267, with a median of five. There were 25 studies with five or less infants with cCD, six with six to 10 infected infants, 11 with 11 to 50 infected infants, four with 50 to 100 infected infants, and one with over 100 infected infants. The percentage of infected infants with clinical signs among all infected infants ranged from 0.0% to 100.0%, with a median of 26.0%. Sixteen studies reported a percentage of 0.0%, five reported 1.0% to 25.0%, 12 reported 26.0% to 50.0%, four reported 51.0% to 99.0%, and seven reported 100.0% of cases with clinical signs. Clinical signs of cCD by study, including their reported frequency, are in **S1 Table**. Eight studies reported infant mortality for cCD, three citing Chagas as cause of death, with other causes being stillbirth, Down's syndrome, congenital cardiopathy, respiratory distress, severe neurological damage, gastroenteritis and dehydration, pneumonia, and secondary septicaemia and pneumococcal meningitis. One study reported four deaths, four studies each reported two infant deaths, and the remaining three each reported one infant death. Time of death ranged from birth to 14 months. Congenital Chagas disease mortality characteristics are in **S2 Table**.

## Primary analyses

Results from the primary analyses can be found in **Table 2**. The primary meta-analysis of the proportion of infants with cCD with clinical signs revealed a pooled proportion of 28.3% (95% CI = 19.0%, 38.5%) infants with cCD that showed clinical signs out of all infants with cCD. This estimate had an $I^2$ inconsistency statistic of 88.6% (95% CI = 86.0%, 90.5%), suggesting considerable heterogeneity between studies for morbidity. The Egger's bias statistic was statistically significant (P < 0.0001), suggesting that publication bias influenced these results. The forest plot and Egger's bias plot can be viewed in **Figs 1 and 2**.

**Table 2. Summary of meta-analysis of pooled proportion of infants with cCD with morbidity and mortality, by subgroup.**

|  | Pooled Proportion % | 95% CI | $I^2$ (%) | 95% CI | Egger's Bias | P-Value* |
|---|---|---|---|---|---|---|
| *Primary Analyses (N = 47)* |  |  |  |  |  |  |
| Morbidity | 28.3 | 19.0,38.5 | 88.6 | 86.0,90.5 | 2.5 | <0.0001 |
| Mortality | 2.2 | 1.3,3.5 | 9.6 | 0.0,37.5 | 0.3 | 0.01 |
| *Subgroup 1 (N = 45)* |  |  |  |  |  |  |
| Infant gold[a] (n = 13) | 18.7 | 6.1,36.1 | 86.8 | 79.2,90.8 | 1.8 | 0.00 |
| Infant other[a] (n = 32) | 32.5 | 23.0,42.9 | 78.9 | 70.5,84.1 | 1.8 | 0.08 |
| *Subgroup 2 (N = 47)* |  |  |  |  |  |  |
| Europe (n = 12) | 20.0 | 11.4,30.2 | 10.2 | 0.0,54.8 | 2.3 | 0.04 |
| Latin America (n = 35) | 29.4 | 18.3,41.8 | 91.3 | 89.3,92.7 | 2.9 | <0.0001 |
| *Subgroup 3 (N = 47)* |  |  |  |  |  |  |
| Hepatosplenomegaly | 12.5 | 6.6,19.9 | 85.8 | 82.2,88.4 | 1.4 | 0.003 |
| Preterm birth | 6.0 | 3.3,9.5 | 61.2 | 44.5,71.1 | 0.8 | 0.0003 |
| Low birth weight | 5.8 | 3.2,9.1 | 60.7 | 43.8,70.8 | 0.7 | 0.0008 |
| Anemia | 4.9 | 2.4,8.2 | 64.1 | 49.2,73.1 | 0.6 | 0.0155 |
| Jaundice | 4.7 | 2.4,7.7 | 59.9 | 42.4,70.3 | 0.4 | 0.0394 |
| *Subgroup 4 (N = 46)* |  |  |  |  |  |  |
| Prior to 2000 | 37.4 | 20.7,55.9 | 94.5 | 93.1,95.4 | 4.0 | 0.0001 |
| 2010–2010 | 21.1 | 8.6,37.2 | 89.8 | 86.2,92.2 | 1.7 | 0.0022 |
| After 2010 | 22.5 | 13.2,33.4 | 29.2 | 0,68.0 | 1.6 | 0.3282 |

*Egger's Bias plot statistical significance for asymmetry

[a]Gold standard diagnosis for infants is defined as confirmation of infection at birth using parasitological examination (microhematocrit or microStrout testing methods), and if negative follow-up parasitological examination if negative, and at 10 months two serological tests

The pooled proportion of infants with cCD that died to all infected infants was 2.2% (95% CI = 1.3%, 3.5%) (**Fig 3**). The $I^2$ inconsistency statistic was 9.6% (95% CI = 0% to 37.5%), suggesting between-study heterogeneity did not influence mortality. The 0.26 Egger's bias statistic (**Fig 4**) was statistically significant (P = 0.0084), suggesting publication bias influenced results.

## Subgroup analyses

Subgroup analysis results are in **Table 2**. Subgroup 1 analysed 45 studies with available information by whether gold standard diagnosis was used for cCD. The pooled proportion of infants with cCD with clinical signs diagnosed with the gold standard was 18.7% (95% CI 6.1%, 36.1%), versus 32.5% (95% CI 23.0%, 42.9%) among infants diagnosed with an alternative.

Subgroup 2 analysed the proportion of infants with clinical signs with cCD by geographic region in 47 studies. European studies (n = 12) had a pooled proportion of 20.0% (95% CI 11.4%, 30.2%) versus 29.4% (95% CI 18.3%, 41.8%) in Latin American studies (n = 35).

Subgroup 3 analysed the proportion of infants with cCD with clinical signs by clinical sign to determine frequency of each clinical sign. Hepatosplenomegaly, reported as either hepatomegaly, splenomegaly, or hepatosplenomegaly, occurred most frequently, with a pooled proportion of 12.5% (95% CI 6.6%,19.9%). The following clinical signs occurred the most frequently after hepatosplenomegaly: preterm birth with a pooled proportion of 6.0% (95% CI 3.3%, 9.5%), low birth weight (LBW) 5.8% (95% CI 3.2%, 9.1%), anemia 4.9% (95% CI 2.4%, 8.2), and jaundice 4.7% (95% CI 2.4%, 7.7%).

Subgroup 4 analysed the proportion of infants with cCD with clinical signs by the year(s) that study data was collected. For studies whose data were collected prior to 2000, the pooled proportion of infants with cCD who showed clinical signs was 37.4% (95% CI 20.7%, 55.9%).

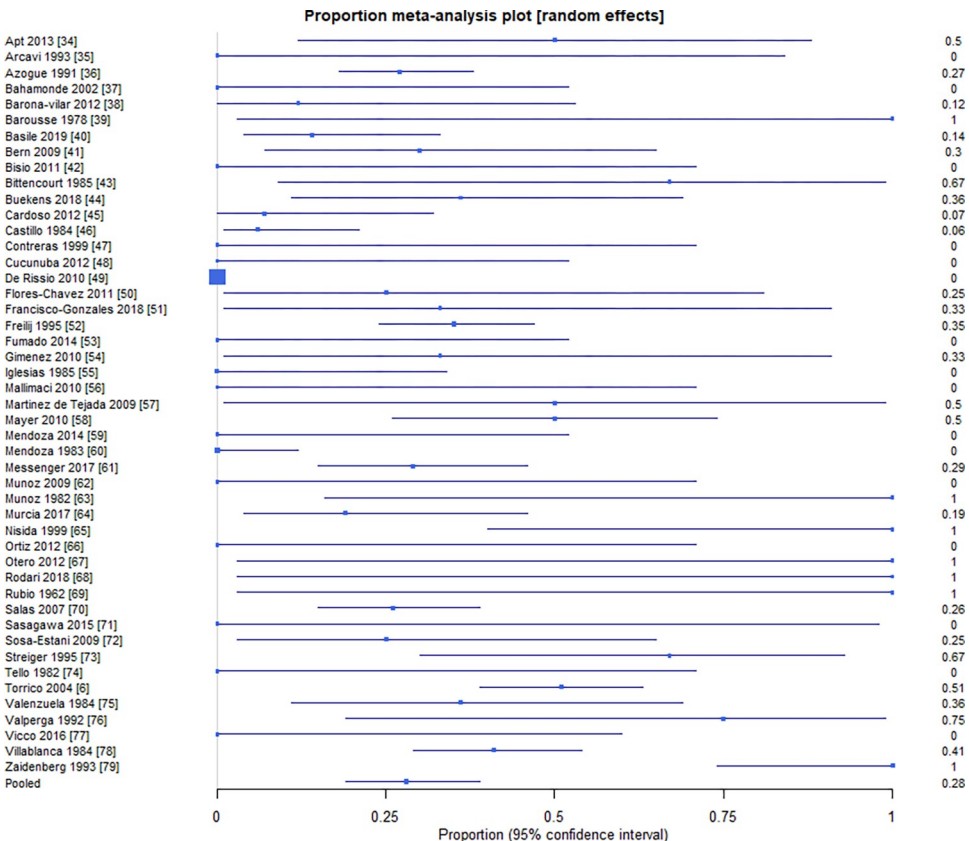

**Fig 1. Proportion of infants with cCD who present with morbidity by study [6,34–79].**

Studies whose data were collected from 2000 to 2010 had a pooled proportion of 21.1% of infants with cCD with clinical signs (95% CI 8.6%, 37.2%), and those who collected data after 2010 showed a pooled proportion of 22.5% (95% CI 13.2%, 33.4%).

## Discussion

### Main findings

Our primary meta-analysis of the proportion of infants with cCD with clinical signs to all infants with cCD revealed a pooled proportion of 28.3% across 47 included studies. The pooled proportion of mortality cases among all infants with cCD was estimated at 2.2%. Sensitivity analyses were conducted to determine robustness of results based on review decisions. Sources of heterogeneity were investigated based on infant characteristics and study characteristics across five subgroups. Detailed results and interpretations are described in **S6 File**.

### Interpretation

Our study expands on the body of work surrounding cCD and to our knowledge is the first to estimate its burden using an exhaustive search strategy that identified 47 studies for meta-analysis. Prior global estimates of the burden of cCD were likely underestimated given the influence of cited issues in diagnosing cCD on population-based data sources [18, 19]. Other estimations have been based on the results of individual observational studies [80, 81]. Our meta-analysis of observational studies allows for a more robust estimation of the burden of

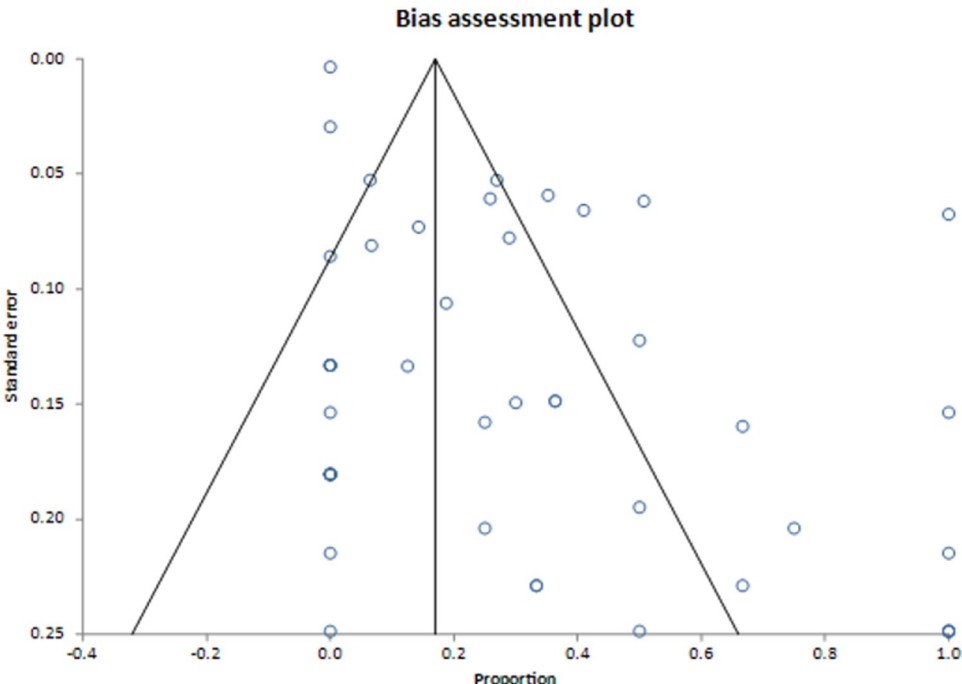

**Fig 2. Bias assessment of meta-analysis of pooled proportion of infants with cCD with morbidity[a].**
[a] Publication bias is considered present when there is asymmetry of the funnel plot.

cCD in comparison to population-based data sources to describe the global burden of Chagas disease. A previous systematic review estimated that the pooled cCD transmission rate was 4.7% (95% CI: 3.9–5.6%) [3]. Our study suggests that of these cCD cases, 28.3% might present with morbidity and 2.2% with mortality. Compared to other congenital infections, about 10.0 to 30.0% of infants with congenital toxoplasmosis present with clinical signs at birth [82] and estimates from a study in Brazil suggest that 11.1% of congenital infections will result in fetal death [83]. In addition, 10.0 to 15.0% of infants born with congenital cytomegalovirus have clinical signs at birth with a mortality rate of <5% [84].

This study has raised concerns about the quality of studies that are conducted on cCD and their ability to attribute clinical signs to the disease. Only two eligible studies compared clinical signs in infected to non-infected mother-infant dyads [6, 61]. Torrico et al. revealed a statistically significant increase in premature rupture of membranes and statistically significant decrease in birth weight and gestational age in infected dyads compared to non-infected dyads [6]. Similarly, Messenger et al. showed that *T.cruzi* infected infants were 2.7 times as likely to be low birthweight compared to non-infected infants (OR = 2.7, 95% CI 1.1, 5.8) [61]. Despite low risk of bias in these two studies, most other included studies were found to be moderate or high risk of bias. Coupled with a lack of comparison group, these studies have limited capability of attributing infected infants' signs to *T. cruzi* infection. Furthermore, cCD morbidity may be influenced by the parasitic load in infected infants and of studies included in our analysis, only one quantified parasitic load [41]. Bern et al., measured the course of parasitic load in infected infants, but did not attribute parasitic load to clinical signs presented in infants [41]. Studies analyzing this association are needed. There is a paucity of quality data on clinical manifestations and outcomes of cCD due to the lack of robust observational studies of cCD and under-resourced country cCD disease control programs. Higher quality research and improved cCD disease control programs are needed.

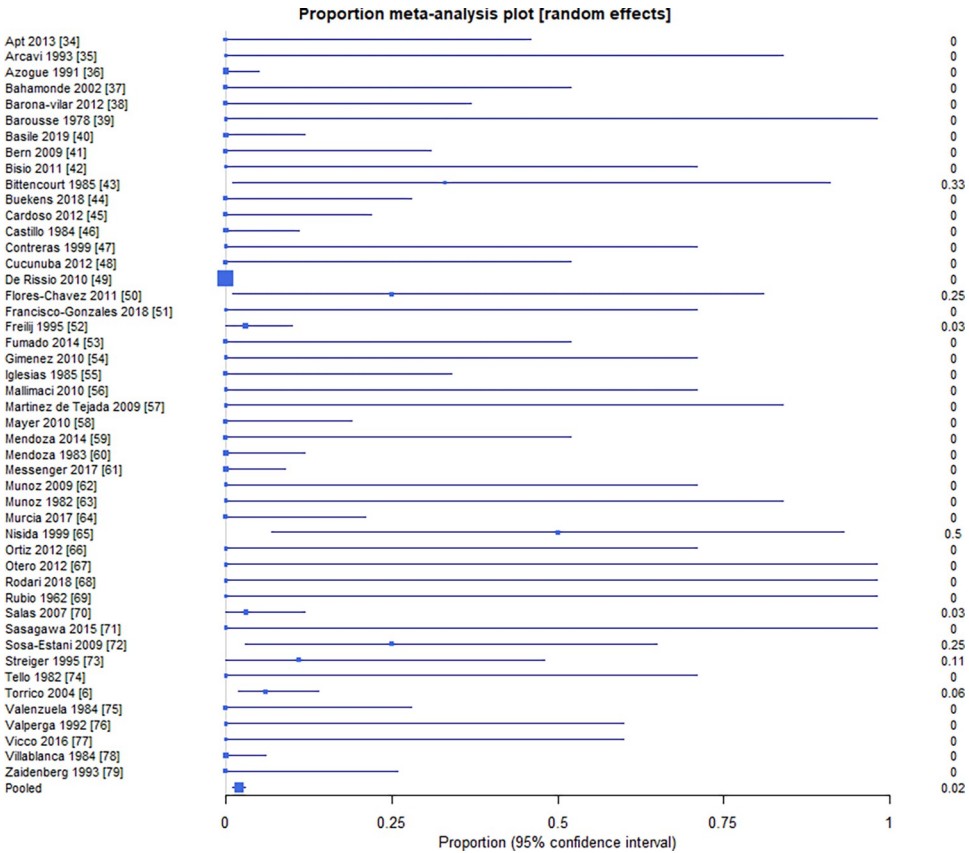

**Fig 3. Proportion of infants with cCD who experience mortality by study [6,34–79].**

There are various barriers to improving quality of cCD research. Chagas disease primarily affects impoverished populations and few resources have been dedicated to addressing the disease, despite the World Health Organization (WHO) defining Chagas disease a neglected tropical disease [85]. Historically, disease control efforts have focused on vector control [86], leaving health systems unprepared to address cCD [87]. This historical lack of emphasis on prevention of cCD has been reflected in the poor quality and paucity of research conducted on its prevention prior to major regional disease control programs in Latin America [88]. In light of this, the body of cCD literature still lacks in quality and further investment is needed.

We identified moderate to high risk of bias in over half of the included studies in reporting of results (71%), exposure and outcome measurement (65%), statistical methods (61%), and declaration of conflict and ethical statements (56%). Studies performed poorly in cCD diagnosis and reporting of these results, which has been cited as an issue due to limited access to and performance of the gold standard diagnostic algorithm, and the subsequent estimated 50% loss to follow-up of at-risk infants [3, 18]. With regard to outcome measurement, some studies only report signs displayed, making it possible some may have been missed if studies did not explicitly evaluate for them. Furthermore, certain clinical signs were not reported frequently enough to be analyzed such as intensive care unit (ICU) admission rate and low Apgar score. Four cases across three studies reported a low Apgar score (below 7 at 1 minute), and seven cases across three studies were admitted to the ICU. Low reporting frequency may be due to limitations in studies method of reporting; however, these clinical signs are an important proxy for clinical severity. Furthermore, the proportion of infants presenting with low birth

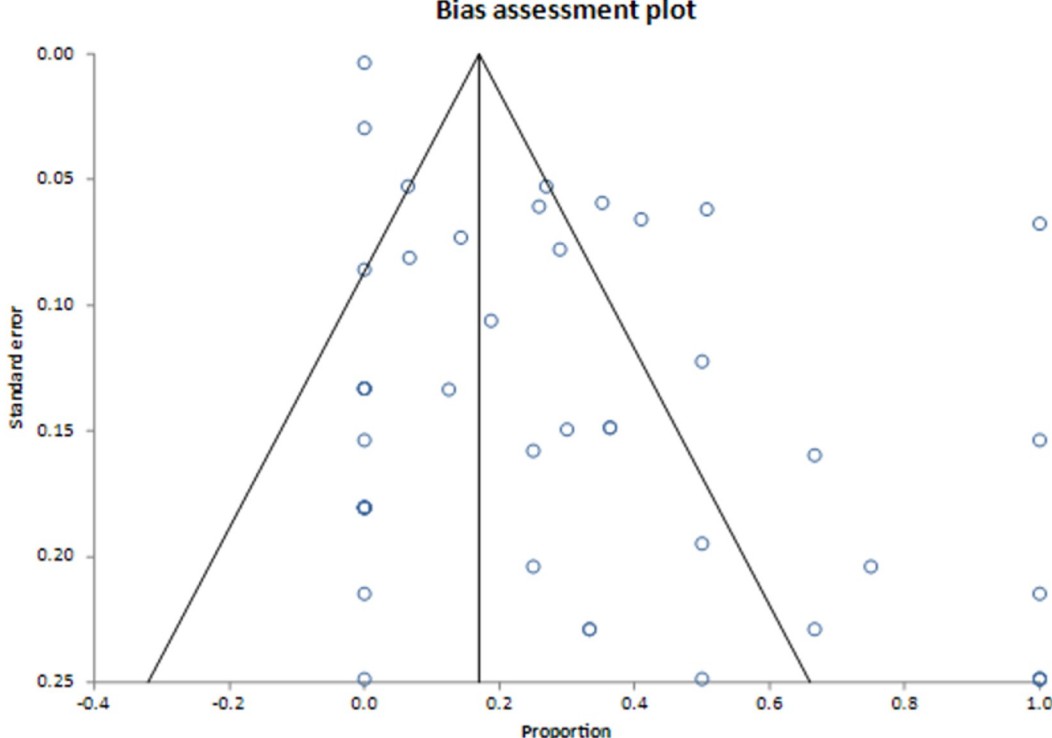

**Fig 4. Bias assessment of meta-analysis of pooled proportion of infants with cCD who experience mortality[a].**
[a] Publication bias is considered present when there is asymmetry of the funnel plot.

weight was 6.0%, lower than the rate in Latin America (8.7%), North America, Europe, Australia, and New Zealand (7.0%), and globally (14.6%) [89]. This number is lower than expected and may be due to issues in outcome measurement and low-quality reporting of results. Given that previous literature has identified signs of cCD through individual studies [2, 6–8, 10, 11, 80], the exhaustive list of signs identified in this study can improve clinical surveillance and guide outcome measurement in future observational research.

The burden of cCD may increase as untreated children grow older and become chronic cases that may develop cardiac and/or gastrointestinal clinical signs [18]. Congenital Chagas disease is almost 100% curable in infants less than 1 year old and treatments are tolerated well [18]. In addition, treating infected women and girls before they bear children can prevent vertical transmission of *T. cruzi* [90,91]. As such, our estimated proportion of 28.3% of cCD cases that present with clinical signs may be preventable through increased screening and treatment. Despite this, an analysis of the 2010 Global Burden of Disease project data revealed that the decrease in Chagas' burden of disease in DALYs was lower than that of other NTDs from 1990 to 2010 [92]. Given this burden is preventable, more investment in disease control and our understanding of its burden is needed.

## Strengths and limitations

This study has several strengths. First, to our knowledge there exists no other study that provides a pooled proportion of infants with cCD with clinical signs. This study employed a comprehensive search strategy, employed on databases that include those primarily focused on Latin American research, without language restrictions. Additionally, estimates produced were precise, as shown by narrow confidence intervals. The subgroup analysis focused on

geographic region allowed for informed analyses of how this factor influences the proportion of infants with cCD with clinical signs. The mortality proportions estimates had low heterogeneity, suggesting studies are similar enough to combine and confidently interpret their results. The subgroup analysis of method of diagnosis informs how using a gold standard diagnosis influences the proportion of infants with cCD with clinical signs. Lastly, subgroup analysis by clinical signs displayed provides further insight on the clinical signs that are indicative of cCD in infants.

This study also has several limitations. First, grey literature was not searched, and given the statistical significance of the Egger's bias estimate, this study is vulnerable to the effects of publication bias and ultimately its generalizability and validity. Additionally, most included studies did not compare morbidity or mortality in infected and non-infected mother-infant dyads. Without the comparison to a non-infected control group, this limits ability to associate signs to cCD. The subgroup analysis of geographic region did not allow for disaggregation of results further than Latin American region due to a small sample size of studies from Mexico and Central America to analyse separately. Furthermore, the subgroup analysis of study date showed differences in results over time and underlying reasons for these differences are unknown and cannot be determined. Certain mortalities such as abortion and stillbirth may be underreported as these cases were only included in this analysis if the fetus has been diagnosed with Chagas disease post-mortem. Apgar scores were only collected at 1 minute as the majority of studies did not report scores at 5 minutes. Additionally, the majority of $I^2$ estimates for morbidity proportions displayed considerable heterogeneity between studies, suggesting inconsistencies between studies are not due to chance alone and thus caution should be used when interpreting results. The risk of bias assessment revealed that overall, 34(72.3%) of included articles had a high risk of bias, 10 (21.3%) of articles had a moderate risk of bias, and only 3(6.4%) of articles had low risk of bias. This, in combination with a significant difference between the sensitivity analysis results excluding those studies with high risk of bias, suggests that the risk of bias influencing the results is high. A post-hoc analysis to determine if an association could be found between *T. cruzi* DTUs and the occurrence of clinical signs in infants with cCD was planned, however, only two studies have performed parasite genotyping, only one of which performed these tests in infants [42,66]. Lastly, there was a large portion of studies with missing data for certain clinical signs and missing values were assumed to be 0. Although this method likely meets the assumption that studies only reported clinical signs that were displayed and all other values were zero, there is a chance this assumption was not met, and bias may have been introduced into these subgroup results due to this imputation.

## Conclusion

Among 47 included studies, the pooled proportion of infections of cCD with clinical signs among all infected fetuses and infants was 28.3%; the pooled proportion of mortality for cCD among all cCD infected fetuses and infants was 2.2%. Caution should be used when interpreting estimated morbidity proportions, as there was considerable heterogeneity between studies. Furthermore, sensitivity analyses revealed that excluding studies with a high risk of bias was significantly lower than the overall proportion (16.6%). Mortality proportions had low heterogeneity between studies and may be interpreted confidently. Studies comparing infected and non-infected mother-infant dyads are needed to determine the morbidity and mortality associated with cCD.

## Supporting information

**S1 File. Morbidity signs of congenital Chagas disease.**
(DOCX)

**S2 File. Search strategy.**
(DOCX)

**S3 File. PRISMA flowchart and hierarchy for consideration of full-text articles.**
(DOCX)

**S4 File. Summary of extracted data.**
(DOCX)

**S5 File. Risk of bias algorithms, summary within-domain risk of bias, and results.**
(DOCX)

**S6 File. Sensitivity analyses results and assessment of heterogeneity.**
(DOCX)

**S1 Dataset. Data extraction form.**
(XLSX)

**S1 Table. Congenital cases morbidity characteristics.**
(DOCX)

**S2 Table. Congenital cases mortality characteristics.**
(DOCX)

## Acknowledgments

The authors would like to thank Agustín Ciapponi (Instituto de Efectividad Clínica y Sanitaria, Buenos Aires, Argentina), Luz Gibbons (Instituto de Efectividad Clínica y Sanitaria, Buenos Aires, Argentina), Eric Dumonteil (Tulane School of Public Health and Tropical Medicine, New Orleans, Louisiana, USA), and Claudia Herrera (Tulane School of Public Health and Tropical Medicine, New Orleans, Louisiana, USA) for their support and advice during this study.

The author's views expressed in this publication do not necessarily reflect the views of their affiliated organizations.

## Author Contributions

**Conceptualization:** Sarah Matthews, Ayzsa Tannis, Karl Philipp Puchner, Maria Elena Bottazzi, Maria Luisa Cafferata, Pierre Buekens.

**Data curation:** Sarah Matthews, Ayzsa Tannis, Karl Philipp Puchner, Maria Elena Bottazzi, Maria Luisa Cafferata, Daniel Comandé, Pierre Buekens.

**Formal analysis:** Sarah Matthews, Ayzsa Tannis, Karl Philipp Puchner, Maria Elena Bottazzi, Maria Luisa Cafferata, Daniel Comandé, Pierre Buekens.

**Investigation:** Sarah Matthews, Ayzsa Tannis, Karl Philipp Puchner, Maria Elena Bottazzi, Maria Luisa Cafferata, Daniel Comandé, Pierre Buekens.

**Methodology:** Sarah Matthews, Ayzsa Tannis, Karl Philipp Puchner, Maria Elena Bottazzi, Maria Luisa Cafferata, Daniel Comandé, Pierre Buekens.

**Writing – original draft:** Sarah Matthews, Ayzsa Tannis.

**Writing – review & editing:** Sarah Matthews, Ayzsa Tannis, Karl Philipp Puchner, Maria Elena Bottazzi, Maria Luisa Cafferata, Daniel Comandé, Pierre Buekens.

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
