## [Decision Letter · Decision Letter 0]

17 May 2022

Dear Tannis,

Thank you very much for submitting your manuscript "Estimation of the morbidity and mortality of congenital Chagas disease: a systematic review and meta-analysis" for consideration at PLOS Neglected Tropical Diseases. As with all papers reviewed by the journal, your manuscript was reviewed by members of the editorial board and by several independent reviewers. In light of the reviews (below this email), we would like to invite the resubmission of a significantly-revised version that takes into account the reviewers' comments. 

We cannot make any decision about publication until we have seen the revised manuscript and your response to the reviewers' comments. Your revised manuscript is also likely to be sent to reviewers for further evaluation.

Sincerely,

Alberto Novaes Ramos Jr

Associate Editor

Guilherme Werneck

Deputy Editor

Reviewer's Responses to Questions

**Key Review Criteria Required for Acceptance?**

**Methods**

-Are the objectives of the study clearly articulated with a clear testable hypothesis stated?

-Is the study design appropriate to address the stated objectives?

-Is the population clearly described and appropriate for the hypothesis being tested?

-Is the sample size sufficient to ensure adequate power to address the hypothesis being tested?

-Were correct statistical analysis used to support conclusions?

-Are there concerns about ethical or regulatory requirements being met?

Reviewer #1: This is a paper reporting the results of a systematic review and metanalysis over a relevant neglected tropical disease, as it is Chagas disease, and over an even more relevant topic such as congenital Chagas disease. The important contribution that is reported in this paper is the proportion of infants diagnosed with congenital Chagas disease at birth presenting with clinical symptoms, which was unknown prior this study. 

Here some recommendations:

Page 3, Line 46: Take note that food can be contaminated by different ways (triatomine feces, crushed triatomines, anal gland secretions from infected marsupials). A more accurate statement could be:

......via oral consumption of contaminated food or beverage, ....

There are several sections in which the following statement is used: cCD infected infants. The correct way to refer to this information would be infants with cCD or congenitally T. cruzi infected children. Please, review this information in Lines 54, 108, 220 (two times), 268-269 (two times), 340, 348

Page 25, Line 384: include the institution and country information per each person being acknowledged.

Reviewer #2: (No Response)

Reviewer #3: This is an interesting systematic review and meta-analysis aiming to identify studies that collected information on infants with congenital Chagas disease. The objective of this systematic review is to determine the morbidity and mortality of cCD and it is clearly stated. The authors have summarized and analyzed 47 eligible studies; the information about the populations and sample sizes is in general well described. However it would be interesting to know how many studies have provided information regarding the phylogenetic lineage or discrete typing unit of the Trypanosoma cruzi populations, and if any association could be found between certain discrete typing units and the occurrence of signs and symptoms of disease. This should be in relation to the geographical distribution of cCD cases with morbidity signs and symptoms among endemic countries (In fact, a high proportion of cCD infants born in European countries are born to mothers infected in Bolivia, where certain parasite genotypes prevail). To analyse the frequency of signs and symptoms according to regions where T.cruzi I, T.cruzi II or T.cruzi V and or T.cruzi VI circulate would be a plus in the analysis. 

Other issue that would be important for congenital Chagas disease morbidity is the parasitic load of the infected infants. It would be interesting to know the amount of such information among the eligible papers.

Reviewer #4: The objective of the study are clearly articulated with testable hypothesis stated. The study design which is a meta-analysis is appropriate and to address the stated objectives. The statistics of the research are clear, understandable and verifiable. There is no concerns about ethical or regulatory requirements.

**Results**

-Does the analysis presented match the analysis plan?

-Are the results clearly and completely presented?

-Are the figures (Tables, Images) of sufficient quality for clarity?

Reviewer #1: The authors performed pooled analysis using several approaches to measure the different outcomes (frequency, proportion) as well as assessment of heterogeneity between studies and sensitivity analysis of results.

Reviewer #2: (No Response)

Reviewer #3: The results are clearly presented and the Tables and Figures are adequate.

Reviewer #4: The results of the study are presented clearly and completely. Tables and figures are of sufficient quality for clarity. The presented analysis matches the analysis plan.

**Conclusions**

-Are the conclusions supported by the data presented?

-Are the limitations of analysis clearly described?

-Do the authors discuss how these data can be helpful to advance our understanding of the topic under study?

-Is public health relevance addressed?

Reviewer #1: The authors acknowledge the limitations of their approach. The conclusions are based on the evidence presented.

Reviewer #2: (No Response)

Reviewer #3: The strengths and weaknesses of the study are clearly expressed in the Discussion. It would be desirable that, al least, any comment about the parasite diversity in relation of cCD transmission and morbidity be taken into account in the discussion.

Reviewer #4: Conclusions of the study are supported by the date presented. The authors discussed and explained how these data can help us better understand the subject under study.

**Editorial and Data Presentation Modifications?**

Reviewer #1: Minor observations

Page 3, Line 28: it is necessary to add the abbreviation after congenital Chagas disease:

…information on infants with congenital Chagas disease (cCD), we summarized…

Do not start a sentence with a number or an abbreviation. If you need to start a sentence with a number or an abbreviation such as a scientific name, present the number in letters and write the complete name. Please review the whole document. Examples of sentences to be reviewed include Lines 43, 49, 188, 194-196, 203, 

Page 5. Line 90: Please review the correct wording: 

In 2010, it was estimated that between…

Page 6, Line 106: Include the PROSPERO registry number in the text.

Page 6, Line 115: To be accurate, include that it refers to congenital infection

T. cruzi congenitally infected infant

Page 21, Line 263: the abbreviation LBW has not been used before in the manuscript. Better to write low birth weight

Page 22, Line 299: review wording: Chagas disease

Page 22, Line 302: the word historically is used two times in the same sentence.

Page 24, Line 354: review wording: the comparison to a non-infected control group

S1 File – Complete the name of the abbreviation ICU 

S1 Table – Complete the name of abbreviations such as NICU, PROM; PTB, MPTB, VPTB, EPTB, LBW, VLBW, ELBW

Review the spelling of the words: anasarca, maxillofacial, pneumopathy, bronchopneumonia, metaphysis

S2 File – Confirm the search term in LILACS: Tiypanosom$

S5 File – Complete the name of abbreviation: Participants SES

Reviewer #2: (No Response)

Reviewer #3: Minor revision: 

Line 90 : Rational . It was estimated instead of its estimated.

Line 142.. Please clarify this part of the sentence : independently assessed included studies’ risk of bias of the included studies

Line 154 variance weight “was” applied to transformed proportions,

Reviewer #4: (No Response)

**Summary and General Comments**

Reviewer #1: This is a paper reporting the results of a systematic review and metanalysis over a relevant neglected tropical disease, as it is Chagas disease, and over an even more relevant topic such as congenital Chagas disease. The important contribution that is reported in this paper is the proportion of infants diagnosed with congenital Chagas disease at birth presenting with clinical symptoms, which was unknown prior this study. The authors performed pooled analysis using several approaches to measure the different outcomes (frequency, proportion) as well as assessment of heterogeneity between studies and sensitivity analysis of results. The authors also acknowledge the limitations of their approach. The conclusions are based on the evidence presented.

Reviewer #2: This manuscript describes the results of a metaanalysis of morbidity and mortality associated with congenitally acquired Chagas disease. The authors have gone to great effort to search the published scientific literature and extract analyzable data from a small subset of the studies they found. As they clearly acknowledge, there are concerns for sources of bias in the analysis but overall their conclusions about manifestations and deaths are informative. They attempted to examine the influence of several factors that might cause heterogeneity but failed to include one potential source, year of publication (better would be year(s) during which the study was performed). Given the wide range of time covered, it is very likely that level of health care, prevalence of comorbid conditions, and other factors may have changed in a given geographic area which would potentially influence outcomes. Especially in the discussion section, the differences between public health disease control programs and research studies is not clear. The lack of better data on clinical manifestations and outcomes of congenital Chagas disease is due to both under resourced country programs and paucity of robust research studies. The advocacy targets and messages for the two are very different. 

Pg. 4 lines 53-63. Although the term is often used in disease descriptions for the general public, in most medical literature symptoms are defined as subjective, as perceived by the patient, which would not pertain to newborn infants. The term can be used to describe a condition that is an indication of an underlying process. This does not seem to be what is described here. 

Pg. 4 lines 69-71. Antibody development may be subsequent to the parasitemia of acute infection. Acute infection is diagnosed by detection of the parasite, either morphologically or by molecular testing (PCR) and patients will have positive results on PCR before they have detectable antibody levels. Diagnosis is clearly described in PAHO 2019 Guidelines for the diagnosis and treatment of Chagas disease and Forsyth et al. J Infect Dis 2022 Recommendations for screening and diagnosis of Chagas disease in the United States. 

Pg. 4 lines 71-73. The currently available rapid diagnostic tests are for antibody detection, please see comment above regarding diagnosis of acute infection. Depending on the setting, rapid tests may be used for as the first test (see PAHO Guidelines). 

Pg. 5 line 79. This is either the Strout method or microStrout method, capitalization is needed.

Pg. 5 lines 81-83. I could not find the evidence to support this statement regarding molecular methods for diagnosis of congenital Chagas disease in the cited reference. 

Pg. 5 line 90. Do you mean it is estimated? Since the cited evidence was from 12 years ago, it would be more accurate to say it was estimated. 

Pg. 6 line 109. The term is randomized controlled trials, no hyphen.

Pg. 6 line 113-114. A study would be designed to collect original data or not and that design would be a criterion for inclusion; this sentence seems out of place in outcomes. Weren’t “articles” the published study results/outcomes? 

Pg. 7 line 125. Do you mean article title? In most published scientific literature, the study that generated the reported results may or may not have had a specific title that would be provided in the written report. At lines 128-129, were the studies duplicated or were the results from the same study published in “duplicate” articles? In the S3 document, articles were duplicates. In some cases, an individual study has had results published more than once, perhaps that was what you meant? Either S3 or the text at lines 128-129 need to be revised to agree.

Pg. 7 line 143 I think a copy and paste error, included studies appears twice.

Pg. 8 line 149. The topics were narratively summarized, not was.

Pg. 8 lines 188-189. Please consider revising to avoid confusion over the term Caribbean region; many readers will interpret that to include the Caribbean islands. 

Pg. 16 line 220. It would be more accurate to compare infants with detected clinical manifestations to those without, or infants with signs to those without.

Pg. 16 Table 2. Please provide a title that orients the reader and better descriptions for column one row titles, even if only with footnotes. The reader has to go back to the methods section to understand what is meant by primary and subgroup analysis and how to interpret “Infant gold” or “Infant other”.

Pgs. 17-20 Figs. 1-4. Please provide descriptive titles and legends for all figures to orient readers. 

Pg. 20 line 257. Please revise, Caribbean is not accurate. 

Pg. 20-21 lines 259-264. Here symptom is used to categorize what was defined as sign on pg. 4. Please revise for consistency and to reflect that infants are not reporting symptoms, but clinical signs may be recognized or diagnosed by the health care provider or caregiver. 

Pg. 22 lines 299-309. This paragraph should be revised or deleted.The WHO designation was intended to draw attention to the neglect, not cause it; Consider revising, perhaps something like this: Chagas disease primarily affects impoverished populations and few resources have been dedicated to addressing the disease, despite the World Health Organization (WHO) defining Chagas disease a neglected tropical disease.

 There is no second mentioned so no need to say “first” (line 299). 

The word historically appears twice in the same sentence, please revise (line 302).

lines 303-304. Not the Caribbean, please correct throughout this manuscript. Why is 2017 in parentheses here? Most studies, poor quality or not, and their published results are not produced as part of ongoing disease control programs. However, consistently funded and supported effective public health programs that cover the jurisdictions needs, as the cited reference advocates, can generate reliable data to monitor disease trends. 

lines 304-306. Consider using the original references that were cited in the AHA position statement overview of Chagas disease. 

lines 308-309. Here the underfunding of disease control and prevention programs are again blamed for a lack of research and studies of Chagas disease. Please revise.

Reviewer #3: The meta-analysis allowed estimation of 28.3% of infants with symptomatic congenital Chagas disease and 2.2% with fatal outcome. Such integrative amount of data about the clinical compromise observed in Congenital Chagas disease infants has not been reported before, so this manuscript can be of true value to the health community working on neglected tropical diseases.

Reviewer #4: (No Response)

PLOS authors have the option to publish the peer review history of their article (what does this mean?). If published, this will include your full peer review and any attached files.

Reviewer #1: No

Reviewer #2: No

Reviewer #3: No

Reviewer #4: Yes: Ali Acar
---

## [Decision Letter · Decision Letter 1]

23 Sep 2022

Dear Tannis,

Thank you very much for submitting your manuscript "Estimation of the morbidity and mortality of congenital Chagas disease: a systematic review and meta-analysis" for consideration at PLOS Neglected Tropical Diseases. As with all papers reviewed by the journal, your manuscript was reviewed by members of the editorial board and by several independent reviewers. The reviewers appreciated the attention to an important topic. Based on the reviews, we are likely to accept this manuscript for publication, providing that you modify the manuscript according to the review recommendations. 

Sincerely,

Alberto Novaes Ramos Jr

Academic Editor

Guilherme Werneck

Section Editor

Reviewer's Responses to Questions

**Key Review Criteria Required for Acceptance?**

**Methods**

-Are the objectives of the study clearly articulated with a clear testable hypothesis stated?

-Is the study design appropriate to address the stated objectives?

-Is the population clearly described and appropriate for the hypothesis being tested?

-Is the sample size sufficient to ensure adequate power to address the hypothesis being tested?

-Were correct statistical analysis used to support conclusions?

-Are there concerns about ethical or regulatory requirements being met?

Reviewer #1: (No Response)

Reviewer #2: No comments.

Reviewer #4: The objectives of the study are clearly stated with a testable hypothesis.

The study design was designed in accordance with the stated objectives.

The population included in the study was clearly defined and suitable for the hypotase tested.

The sample size included in the study is sufficient to provide sufficient test power to address the hypothesis being tested.

 The statistical analyzes used are appropriate and accurate.

There is no ethical concern.

**Results**

-Does the analysis presented match the analysis plan?

-Are the results clearly and completely presented?

-Are the figures (Tables, Images) of sufficient quality for clarity?

Reviewer #1: (No Response)

Reviewer #2: No comments.

Reviewer #4: Analyzes were prepared in accordance with the analysis methods.

The results are clear and straightforward.

Tables are adequate and of sufficient quality.

**Conclusions**

-Are the conclusions supported by the data presented?

-Are the limitations of analysis clearly described?

-Do the authors discuss how these data can be helpful to advance our understanding of the topic under study?

-Is public health relevance addressed?

Reviewer #1: (No Response)

Reviewer #2: Thank you for including time (year of data collection/publication) in your analyses. However, the interpretation of the differences you found is concerning. Lines 403-404. The differences in studies’ results over time are a limitation of this metanalysis, not a strength. The causes for those differences are undefined, you cannot assume they are due to “trends over time in cCD management and control”. Unless there is evidence to support the assumption, this should be moved to the limitations section and state that the underlying reasons for the differences between time categories is undefined/unknown.

Reviewer #4: The results support the data.

The method and limitations of the analyzes are clearly stated.

The authors emphasized how the data would assist in understanding the topic.

It is a study on public health and will contribute scientifically in this field.

**Editorial and Data Presentation Modifications?**

Reviewer #1: Minor revision

The authors completed to satisfaction all the recommendations, except the following:

Line 46-47: should say: …….via oral consumption of contaminated food or beverage [ELIMINATE THIS "with triatomine feces"], and through vertical transmission from mother....Note: This is because food and beverages can be contaminated by other than triatomine feces, for instance crushed infected insects and anal secretions from infected marsupials.

Line 108-109: should say: ……….associated with cCD [ELIMINATE THIS: "infection"] were considered…..Note: This is because is congenital Chagas disease or T. cruzi congenital infection but not congenital Chagas disease infection

Reviewer #2: As above, a revision to limitations is needed.

Reviewer #4: Accept

**Summary and General Comments**

Reviewer #1: (No Response)

Reviewer #2: No additional comments to this revision.

Reviewer #4: (No Response)

PLOS authors have the option to publish the peer review history of their article (what does this mean?). If published, this will include your full peer review and any attached files.

Reviewer #1: Yes: Jackeline Alger

Reviewer #2: No

Reviewer #4: Yes: Ali Acar

Figure Files:

Data Requirements:

Reproducibility:

References

---

## [Decision Letter · Decision Letter 2]

21 Oct 2022

Dear Tannis,

We are pleased to inform you that your manuscript 'Estimation of the morbidity and mortality of congenital Chagas disease: a systematic review and meta-analysis' has been provisionally accepted for publication in PLOS Neglected Tropical Diseases.

Best regards,

Alberto Novaes Ramos Jr

Academic Editor

Guilherme Werneck

Section Editor

Reviewer's Responses to Questions

**Key Review Criteria Required for Acceptance?**

**Methods**

-Are the objectives of the study clearly articulated with a clear testable hypothesis stated?

-Is the study design appropriate to address the stated objectives?

-Is the population clearly described and appropriate for the hypothesis being tested?

-Is the sample size sufficient to ensure adequate power to address the hypothesis being tested?

-Were correct statistical analysis used to support conclusions?

-Are there concerns about ethical or regulatory requirements being met?

Reviewer #1: (No Response)

Reviewer #2: (No Response)

Reviewer #4: The objectives of the study are clearly stated with a testable hypothesis.

The study design was designed in accordance with the stated objectives.

The population included in the study was clearly defined and suitable for the hypotase tested.

The sample size included in the study is sufficient to provide sufficient test power to address the hypothesis being tested.

The statistical analyzes used are appropriate and accurate.

There is no ethical concern.

**Results**

-Does the analysis presented match the analysis plan?

-Are the results clearly and completely presented?

-Are the figures (Tables, Images) of sufficient quality for clarity?

Reviewer #1: (No Response)

Reviewer #2: (No Response)

Reviewer #4: Analyzes were prepared in accordance with the analysis methods.

The results are clear and straightforward.

Tables are adequate and of sufficient quality.

**Conclusions**

-Are the conclusions supported by the data presented?

-Are the limitations of analysis clearly described?

-Do the authors discuss how these data can be helpful to advance our understanding of the topic under study?

-Is public health relevance addressed?

Reviewer #1: (No Response)

Reviewer #2: (No Response)

Reviewer #4: The results support the data.

The method and limitations of the analyzes are clearly stated.

The authors emphasized how the data would assist in understanding the topic.

It is a study on public health and will contribute scientifically in this field.

**Editorial and Data Presentation Modifications?**

Reviewer #1: (No Response)

Reviewer #2: (No Response)

Reviewer #4: Accept

**Summary and General Comments**

Reviewer #1: (No Response)

Reviewer #2: Thank you for the revision to move the influence of time to the limitations section.

Reviewer #4: (No Response)

PLOS authors have the option to publish the peer review history of their article (what does this mean?). If published, this will include your full peer review and any attached files.

Reviewer #1: **Yes: **Jackeline Alger

Reviewer #2: No

Reviewer #4: **Yes: **Ali Acar

---

## [Editor Report · Acceptance letter]

3 Nov 2022

Dear Tannis,

We are delighted to inform you that your manuscript, "Estimation of the morbidity and mortality of congenital Chagas disease: a systematic review and meta-analysis," has been formally accepted for publication in PLOS Neglected Tropical Diseases.

Best regards,

Shaden Kamhawi

co-Editor-in-Chief

Paul Brindley

co-Editor-in-Chief
